# PROPAGATION TO THE UPPER ATMOSPHERE OF ACOUSTIC-GRAVITY WAVES FROM ATMOSPHERIC FRONTS IN THE MOSCOW REGION

Yuliya Kurdyaeva[1,2], Sergey Kulichkov[3,4], Sergey Kshevetskii[1], Olga Borchevkina[1,2], and Elena Golikova[3]

[1]I.Kant Baltic Federal University, Kaliningrad, Russia
[2]Kaliningrad Branch of the Institute of Earth Magnetism, Ionosphere and Radio Wave Propagation RAS, Kaliningrad, Russia
[3]A.M. Obuckhov Institute of Atmospheric Physics RAS, Moscow, Russia
[4]Moscow State University, Moscow, Russia

**Correspondence:** Yuliya Kurdyaeva (yakurdyaeva@gmail.com)

**Abstract.** The paper uses experimental data of pressure variations on the Earth's surface during the passage of an atmospheric front recorded by a network of 4 microbarographs in the Moscow region. Applying this experimental data, empirical approximations of atmospheric pressure field oscillations are suggested. The obtained approximating surface pressure functions are used as the lower boundary condition for simulating the vertical propagation of acoustic-gravity waves from a source in the lower troposphere. Estimates of the amplitude of temperature disturbances in the upper atmosphere caused by acoustic-gravity waves from a propagating atmospheric front are obtained. For the amplitude of wave temperature disturbances, values of about 200 K are obtained. The amplitude of temperature disturbances in the upper atmosphere caused by background pressure fluctuations on the Earth's surface is estimated at 4–5 K.

**Key-words:** atmosphere, numerical simulation, acoustic-gravity waves, upper atmosphere, atmospheric front.

## 1 Introduction

Changes in the parameters of the upper atmosphere and ionosphere are often associated with meteorological phenomena (Blanc et al. (2014)), which generate acoustic-gravity waves (AGWs). When reaching the altitudes of the upper atmosphere and the ionosphere and dissipating, AGWs can change the state of the atmosphere (Pierce and Coroniti (1966)). AGWs in the lower atmosphere can be generated by atmospheric fronts, jet streams (Ploogonven and Snyder (2007), Ploogonven and Zhang (2014)) with maximum efficiency at altitudes of 9-12 km (Medvedev and Gavrlov (1995)) and mesoscale turbulence (Fritts and Alexander (2003), Fritts et al. (2006)). Atmospheric waves are also often generated by atmospheric convection powered fed by heating and cooling of gas during phase transitions of water (Blanc et al. (2014), Pierce and Coroniti (1966), Balachandran (1980), Alexander et al. (2004), Miller (1999), Fovell et al. (1992)).

AGWs dissipating in the upper atmosphere can cause jet flows and affect the heat balance. Atmospheric waves coming to the altitudes of the ionosphere, affect the ionospheric plasma movement and, as a consequence, the conditions of radio wave propagation.

Numerical simulation of AGW propagation is an effective tool of studying dynamic processes in the atmosphere. One of the difficulties of simulating waves from meteorological sources is that these sources are very diverse and have a complex spatial structure evolving in time. The available experimental information is usually insufficient for a realistic detailed description of such wave sources. The authors noted in (Miller (1999), Fovell et al. (1992), Snively and Pasko (2003)) that meteorological sources excite short-period acoustic-gravity waves; the spectrum and the space-time pattern of the simulated wave process are in good agreement with the observations. However, based on numerical simulations it is difficult to estimate the amplitudes of the generated AGWs due to the lack of detailed information on tropospheric sources (Kshevetskii and Gavrilov (2005)).

The uncertainty in the parameters of tropospheric wave sources significantly affects the accuracy and reliability of the results obtained. Estimation of the amplitudes of waves propagating to the upper atmosphere from tropospheric sources is necessary to study these waves. In addition, knowledge of the wave amplitudes is important for understanding the influence of these waves on the parameters of the upper atmosphere at various altitude levels. Phase transitions of water in the atmosphere are accompanied by the release/absorption of heat during the formation and evolution of clouds and alter the atmospheric pressure. Changes in atmospheric pressure during the formation and evolution of meteorological processes are recorded by microbarographs. These registered forms of wave variations in atmospheric pressure can be used to develop models of acoustic-gravity wave sources. The generated waves can propagate to the upper atmosphere.

The problem of acoustic-gravity wave propagation from variations in density and temperature on the Earth's surface was studied mathematically in Kurdyaeva et al. (2018). The study showed that the variable pressure on the Earth's surface uniquely determines the wave pattern, which however does not depend on the details of the temperature and density dynamics on the Earth's surface (Kurdyaeva et al. (2018)). A numerical model of wave propagation from pressure variations on the Earth's surface was developed in Kurdyaeva et al. (2018). The problem of wave propagation from pressure variations set at the lower boundary is solved analytically in the case of an isothermal atmosphere. A test comparison of numerical and analytical solutions showed that the model (Kurdyaeva et al. (2018)) gives a very good agreement between the numerical solution and the analytical results. The numerical model was also used in Kshevetskii (2001c), Kshevetskii (2001a), Kshevetskii (2002), Kshevetskii (2001b).

## 2   EXPERIMENTAL DATA ON ATMOSPHERIC PRESSURE VARIATIONS

In this study, we use the data of observations of atmospheric pressure variations in 2016 obtained on 4 microbarographs of the A.M. Obukhov Institute of Atmospheric Physics RAS. All microbarographs are located in the Moscow region (the points are at the Moscow State University, MosRentgen, A.M. Obukhov Institute of Atmospheric Physics, RAS, and Zvenigorod Research Station (ZRS) of the A.M. Obukhov Institute of Atmospheric Physics, RAS, Fig.1 (Kulichkov et al. (2017)). The microbarographs record variations in atmospheric pressure in the frequency range from $10^{-4}$ Hz to 3 Hz. We process the data for 2016 and highlight the cases when the amplitude of pressure variations significantly exceeded the background variations. At some moments on July 17-18, 2016, the amplitude of pressure variations exceeded the average 30 times. To perform numerical simulations, we take pressure variation data for July 17-18, 2016.

Below we present the results of simulations of the propagation of atmospheric waves from surface pressure variations obtained from observational data. No similar simulations have been previously performed in the existing literature.

Since we need to estimate the range of amplitudes of the generated waves, we take the case of extreme variations in surface atmospheric pressure when we construct the AGW source. Performing numerical simulations for cases of extreme values of the amplitudes of pressure oscillations on the Earth's surface allows neglecting the constantly existing background wave oscillations in the atmosphere caused by various other wave sources. The change in atmospheric pressure can be associated with the activity of meteorological sources. However, it is impossible to completely cancel the effects of other wave sources unrelated to the meteorological events under consideration on the wave pattern in the upper atmosphere, for example, of sources of oscillations that are anthropogenic in nature. The study of wave propagation from the observed extreme pressure fluctuations allows hoping that the possible influence of various other wave sources on the wave pattern,which is undesirable for our purposes, is leveled. This increases the reliability of calculations. Graphs of pressure variations recorded by four microbarographs on July 18, 2016 are shown in Fig.2.

We use observational data available in the MERRA-2 database (https://disc.gsfc.nasa.gov/) to analyze the regional meteorological situation in the lower atmosphere in the period under study. The surface pressure maps for July 17-18, 2016 obtained from the evaluation of the reanalysis archive data are shown in Fig.3 Moscow $(55°45N, 37°37E)$ is marked in the map with an asterisk. The data of MERRA-2 are presented over the same horizontal grid. The grid has 576 points in the longitudinal direction and 361 points in the latitudinal direction, which corresponds to the resolution of $0.625°×0.5°$.

Fig.3 shows a gradual change in atmospheric pressure. The emerging region of low pressure is due to the formation of a cyclone. The low pressure area extends in the direction of Moscow and the Moscow region. The arrival of a cyclone is usually accompanied by intense cloud formation (Pogosyan (1976)). Thus, the pressure variations registered by microbarographs are likely due to the weather phenomena observed at this time. We believe that other wave sources similar in energy release can be excluded from consideration.

## 3 MATHEMATICAL MODEL

The three-dimensional supercomputer model "AtmoSym" (http://atmos.kantiana.ru/), developed by S.P. Kshevetsky and N.M. Gavrilov (Kshevetskii (2001c), Kshevetskii (2001a), Kshevetskii (2002),Kshevetskii (2001b), Kshevetskii and Gavrilov (2005)) is used to model the propagation of waves upward from pressure variations near the Earth's surface. The model uses parallel computing (Sadovnichy et al. (2013)) and it allows solving a wide range of problems of wave propagation from various initial disturbances and wave sources within the altitude range of 0-500 km over the territory with a horizontal scale of up to several thousand kilometers. The model is adapted in Kshevetskii (2001c) for solving the problems on vertical propagation of waves from pressure variations on the Earth's surface.

The «AtmoSym» numerical model is based on solving a complete system of nonlinear hydrodynamic equations (1) for a gas in a gravity field:

$$\frac{\partial \rho}{\partial t} + \frac{\partial \rho u}{\partial x} + \frac{\partial \rho v}{\partial y} + \frac{\partial \rho w}{\partial z} = 0, \tag{1}$$

$$\frac{\partial \rho u}{\partial t} + \frac{\partial \rho u^2}{\partial x} + \frac{\partial \rho uv}{\partial y} + \frac{\partial \rho uw}{\partial z} = -\frac{\partial p}{\partial x} + \left(\frac{\partial^2}{\partial x^2} + \frac{\partial^2}{\partial y^2}\right)\zeta(z)\,u + \frac{\partial}{\partial z}\zeta(z)\frac{\partial}{\partial z}u,$$

$$\frac{\partial \rho v}{\partial t} + \frac{\partial \rho uv}{\partial x} + \frac{\partial \rho v^2}{\partial y} + \frac{\partial \rho vw}{\partial z} = -\frac{\partial p}{\partial y} + \left(\frac{\partial^2}{\partial x^2} + \frac{\partial^2}{\partial y^2}\right)\zeta(z)\,v + \frac{\partial}{\partial z}\zeta(z)\frac{\partial}{\partial z}v,$$

$$\frac{\partial \rho w}{\partial t} + \frac{\partial \rho uw}{\partial x} + \frac{\partial \rho vw}{\partial y} + \frac{\partial \rho w^2}{\partial z} = -\frac{\partial p}{\partial z} - \rho g + \left(\frac{\partial^2}{\partial x^2} + \frac{\partial^2}{\partial y^2}\right)\zeta(z)\,w + \frac{\partial}{\partial z}\zeta(z)\frac{\partial}{\partial z}w,$$

$$\frac{1}{\gamma - 1}\left(\frac{\partial P}{\partial t} + \frac{\partial Pu}{\partial x} + \frac{\partial Pv}{\partial y} + \frac{\partial Pw}{\partial z}\right) = -P\left(\frac{\partial u}{\partial x} + \frac{\partial v}{\partial y} + \frac{\partial w}{\partial z}\right) +$$

$$+ \left(\frac{\partial^2}{\partial x^2} + \frac{\partial^2}{\partial y^2}\right)\kappa(z)T + \frac{\partial}{\partial z}\kappa(z)\frac{\partial}{\partial z}T + Q_0(z) + Q_{viscous}$$

$$Q_0(z) = -\frac{\partial}{\partial z}\kappa(z)\frac{\partial}{\partial z}T_0(z),\, P = \frac{\rho RT}{\mu}$$

$$Q_{viscous} = \zeta(z)\left(\left(\frac{\partial u}{\partial x}\right)^2 + \left(\frac{\partial u}{\partial y}\right)^2 + \left(\frac{\partial u}{\partial z}\right)^2 + \left(\frac{\partial v}{\partial x}\right)^2 + \left(\frac{\partial v}{\partial y}\right)^2 + \left(\frac{\partial v}{\partial z}\right)^2 + \right. \tag{2}$$

$$\left. \left(\frac{\partial w}{\partial x}\right)^2 + \left(\frac{\partial w}{\partial y}\right)^2 + \left(\frac{\partial w}{\partial z}\right)^2\right)$$

where $t$ is time; $p, \rho, T$ are pressure, density, and temperature; $Rg$ is the universal gas constant; $x, y, z$ and $u, v, w$ are the coordinates and velocity components, respectively; $\gamma$ is the adiabatic constant; $\mu$ is the molecular weight; $g$ is the gravity acceleration; $\zeta$ and $\kappa$ are viscosity and thermal conductivity coefficients; $T_0(z)$ is the background temperature profile. $Q_{viscous}$ is the force of viscous friction.

The dependences of medium parameters (viscosity and thermal conductivity coefficients, background density, temperature and pressure) on altitude are taken from the empirical model of the atmosphere NRLMSISE-00 (Picone et al. (2002)). The vertical grid is uneven; the optimal vertical grid is constructed by the program based on the real medium stratification.

The formulation of the problem of wave generation by pressure variations at the lower boundary is as follows. The system of equations (1) describes wave propagation. The periodic conditions at the horizontal boundaries of the computational domain are applied:

$$\begin{aligned}
u(x = L_x, y, z, t) &= u(x = 0, y, z, t), & u(x, y = L_y, z, t) &= u(x, y = 0, z, t), \\
v(x = L_x, y, z, t) &= v(x = 0, y, z, t), & v(x, y = L_y, z, t) &= v(x, y = 0, z, t), \\
w(x = L_x, y, z, t) &= w(x = 0, y, z, t), & w(x, y = L_y, z, t) &= w(x, y = 0, z, t), \\
\rho(x = L_x, y, z, t) &= \rho(x = 0, y, z, t), & \rho(x, y = L_y, z, t) &= \rho(x, y = 0, z, t), \\
p(x = L_x, y, z, t) &= p(x = 0, y, z, t), & p(x, y = L_y, z, t) &= p(x, y = 0, z, t), \\
T(x = L_x, y, z, t) &= T(x = 0, y, z, t), & T(x, y = L_y, z, t) &= T(x, y = 0, z, t).
\end{aligned} \tag{3}$$

Periodic boundary conditions have the following disadvantage:with this formulation of the problem, the wave leaving a computational domain, for example, through the left boundary of the computational domain, then enters through the right boundary. Nevertheless, in this case, the distance travelled by the wave gradually increases and the amplitude of the wave propagating from the source gradually decreases due to spherical divergence and, cylindrical divergence at lager timeframes. Therefore, given sufficiently large dimensions of the computational domain, the influence of the finiteness of the computational domain on the wave characteristics of interest, such as wave frequencies, spatial scales, amplitude, is not strong. The size of the computational domain in the horizontal plane is chosen experimentally. In this paper, for comparison, simulations are performed for two different computational domains in horizontal of 1020 km × 1020 km and 1320 km × 1320 km and the results showed that periodic conditions do not have a strong effect on the wave parameters of interest due to large computational domain sizes.

Since we are interested only in waves generated by pressure fluctuations at the Earth's surface, the initial conditions

$$u(x,z,t=0)=0, \qquad w(x,z,t=0)=0, \qquad v(x,z,t=0)=0, \qquad \rho(x,z,t=0)=\rho_0(z), \qquad T(x,z,t=0)=T_0(z) \quad (4)$$

correspond to the wave absence at the initial moment in time.

The upper boundary is at the altitude of h=500 km, and the upper boundary conditions are traditional for models of the thermosphere:

$$\frac{\partial T}{\partial z}(x,y,z=h,t)=0, \qquad \frac{\partial u}{\partial z}(x,y,z=h,t)=0, \qquad \frac{\partial v}{\partial z}(x,y,z=h,t)=0, \qquad w(x,y,z=h,t)=0. \tag{5}$$

The conditions at the bottom are special:

$$u(x,y,z=0,t)=0, \qquad v(x,y,z=0,t)=0, \qquad \frac{\partial w(x,y,z=0,t)}{\partial z}=0,$$
$$T(x,y,z=0,t)=T_0(0), \qquad P(x,y,z=0,t)=P_0(0)+f_p(x,y,t), \tag{6}$$

where $f_p(x,y,t)$ is a function describing the wave variations of the pressure field determined empirically based on experimental observations, and $P_0(0)$ is the pressure on the Earth's surface.

## 4 APPROXIMATION OF THE PRESSURE FIELD VARIATIONS ON THE EARTH'S SURFACE NEAR MICROBAROGAPHS

The behavior of the pressure field variations in the vicinity of each of the 4 microbarographs is modeled by the function:

$$f_{p,i}(x,y,t)=\exp\left(-\frac{(x-x_i)^2+(y-y_i)^2}{\lambda^2}\right)q_i(t). \tag{7}$$

Here, the values $(x_i,y_i)$ are the coordinates of the microbarograph with number $i$, and each function $q_i(t)$ describes the behavior of the wave additive to the background pressure on the $i$-th microbarograph. The functions $q_i(t)$ are obtained by interpolation of the real atmospheric pressure digitized with a 12-second step. The $\lambda$ parameter characterizes the effective width of the boundary source and is determined empirically, based on a study of the correlation of the readings of microbarographs,

depending on the distance between them. If the microbarographs are located quite close, then the simulations do not critically depend on the value of the $\lambda$ parameter. The resulting field of variations in atmospheric pressure is obtained by adding the individual fields $f_{p,i}(x,y,t)$ corresponding to the pressure variations in the vicinity of each microbarograph:

$$f_p(x,y,t) = \sum_{i=1}^{4} f_{p,i}(x,y,t)\eta(t)\left(1 - \exp\left(-\frac{t}{\tau}\right)\right) \tag{8}$$

However, if $\sum_{i=1}^{4} \exp\left(-\frac{(x-x_i)^2 + (y-y_i)^2}{\lambda^2}\right) > 1$ is realized at some points on the Earth's surface, then at these points the function $f_p(x,y,t)$ from (8) is divided by $\sum_{i=1}^{4} \exp\left(-\frac{(x-x_i)^2 + (y-y_i)^2}{\lambda^2}\right) > 1$. This is due to the fact that in this case the approximating fields $f_p(x,y,t)$ introduced in the vicinity of each microbarograph overlap.

The Heaviside function is introduced in (8) to turn on the source at $t = 0$, and the multiplier $\left(1 - \exp\left(-\frac{t}{\tau}\right)\right)$ is entered to remove transient effects due to abrupt switching on of the source; $\tau = 300$ sec.

## 5 RESULTS OF SIMULATION OF THE VERTICAL WAVE PROPAGATION FROM ATMOSPHERIC PRESSURE VARIATIONS

In this work, for the control of reliability, simulations are performed for two computational regions, which horizontally are 1020 km × 1020 km and 1320 km × 1320 km.

The temperature field 7 minutes after the source is "turned on" is shown in Fig.4a, 4d. Here the wave amplitudes are still
small. Waves are generated by a combination of four boundary sources, and the wave field near the sources is asymmetrical. However, as the waves are farher from the sources, the waves produced by different interfere pattern in such a way that the wave field looks like a wave field from a single point source. It is explained by the fact that the source centers at the lower boundary, corresponding to different microbarographs are close to each other. The distance between the sources is on average 15 km and does not exceed 50 km, while the curvature radius of the wave front in the simulations in Fig.4a is about 150
20 km, which is significantly longer than the distances between the individual sources. Therefore, the combination of individual sources manifests itself at a far distance as a single point source. By the wave pattern, one can assume that the source is located at a certain altitude. However, the waves actually propagate from the Earth's surface. In Fig.4 we observe mainly acoustic waves and short internal gravity waves.

Fig.4b, e show the wave field 40 minutes and in Fig.4c, f 55 minutes after the switching on of the boundary source, re-
25 spectively, for different areas. The wave pattern becomes developed, but it is mainly formed by acoustic waves and short internal gravity waves propagating from the source on the Earth's surface, as in the previous Fig.4a, d. The wave pattern is symmetrical and the waves propagate as if from a point source. This observation may be useful, since it suggests the possibility of replacing a wave-source having a complex spatial structure with a point wave source. Then one can significantly reduce the amount of computation by replacing the overall three-dimensional model with a much more efficient three-dimensional
centrally symmetric model.

In Fig.4, one can see that during the first hour the amplitude of the wave temperature fluctuations gradually increases and reaches 40 K for the region 1020 km x 1020 km and 50 K for the region 1320 km × 1320 km.

The finiteness of the computational domain does not have a significant effect on the wave parameters of interest, since the amplitude of the waves arriving in the high atmosphere increases with time, whereas the previously formed waves gradually disappear.

Fig.5 shows the temperature perturbation field after 1 hour 30 minutes, 2 hours, 2 hours 30 minutes after switching on of the source. The amplitude of the temperature wave fluctuations gradually increases to 200K. There are no significant differences in wave parameters between the domains. The waves arising from pressure fluctuations near the Earth surface have significantly different scales. Small perturbations have a half-wave size of about 50 km. For a perturbation of the largest scale, the half-wave size is approximately 600 km. The main heating of the medium by waves takes place at altitudes of 100–200 km and the horizontal structure of the perturbation at these altitudes varies significantly. Heating of the medium is also significant at altitudes above 300 km, at horizontal distances of up to 250-300 km from the center of the source.

## 6 Conclusions

The data of wave pressure variations for 2016 in the Moscow region has been processed. These data are used to estimate the amplitude of wave disturbances in the upper atmosphere, caused by experimentally observed wave oscillations of pressure on the Earth's surface.

To estimate the amplitude range of the generated waves, an extreme case is chosen for simulations, when the atmospheric pressure fluctuations were 30 times higher than the average. This event took place from about 8 pm to 11 pm local time on July 17, 2016 and was due to the approaching cyclone.

The field of pressure variations on the ground in the neighborhood of microbarographs has been approximated based on the measurement data from 4 microbarographs. The three-dimensional hydrodynamic initial-boundary problem of the generation of waves by wave-like pressure variations at the lower boundary is solved and the wave field in the upper atmosphere resulting from the observed pressure variations on the Earth's surface has been simulated.

It is found that the amplitude of temperature wave disturbances excited in the upper atmosphere can reach about T = 200K. The real simulation time does not exceed 3 hours, and the wave disturbances in the upper atmosphere can be attributed to infrasonic and relatively high-frequency internal gravity waves (during the simulation time, low-frequency internal gravity waves do not have time to propagate to altitudes higher than 120 km). A comparison of the considered case of extreme pressure fluctuations with the average wave pressure oscillations on the Earth's surface gives an estimate of the amplitude of typical temperature fluctuations due to the propagation of acoustic-gravity waves from below approximately at T = 4-5 K.

*Acknowledgements.* The research is partially supported by the Russian Foundation for Basic Research, projects No. 17-05-00574. The research is carried out using the equipment of the shared research facilities of HPC computing resources at Lomonosov Moscow State University.

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

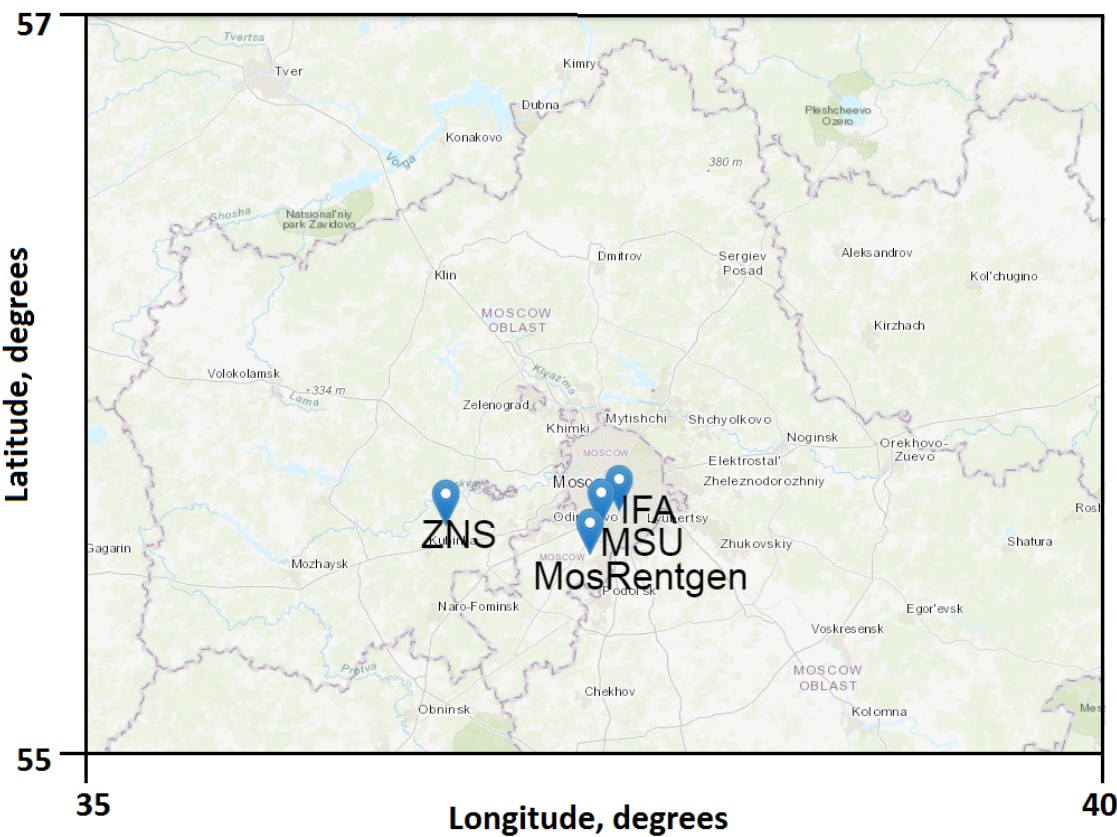

**Figure 1.** Location of microbarographs in the Moscow region.

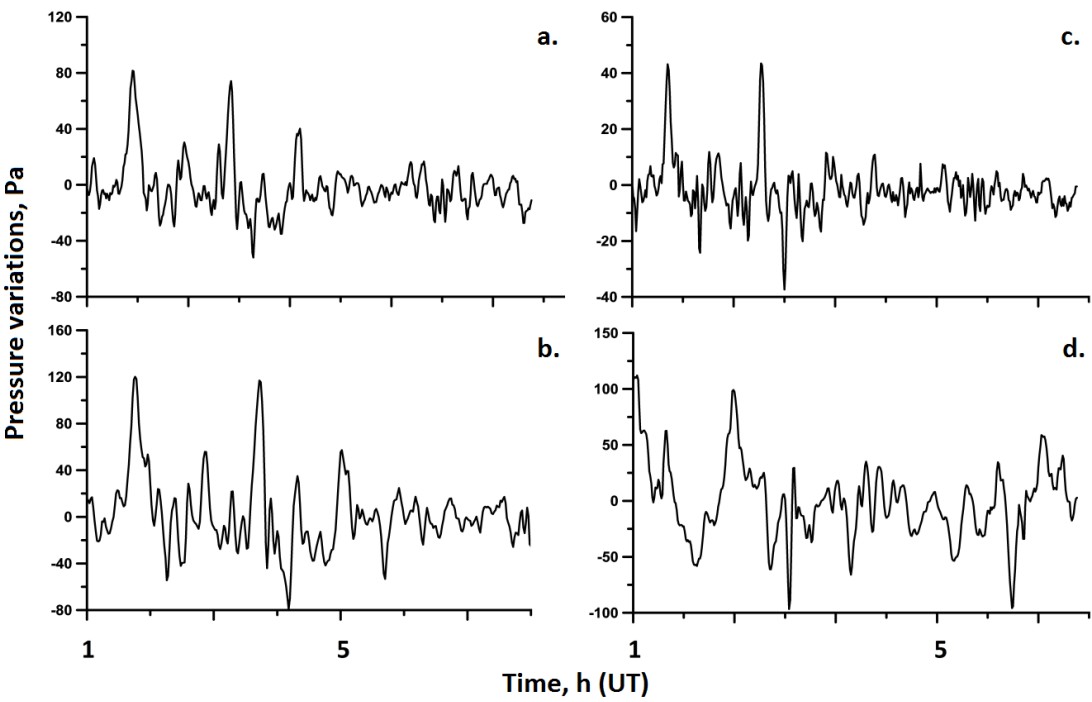

**Figure 2.** Pressure change at 4 stations in Moscow and environs July 18, 2016: a –IFA, b - Moscow State University,  - MosRentgen, d - ZRS.

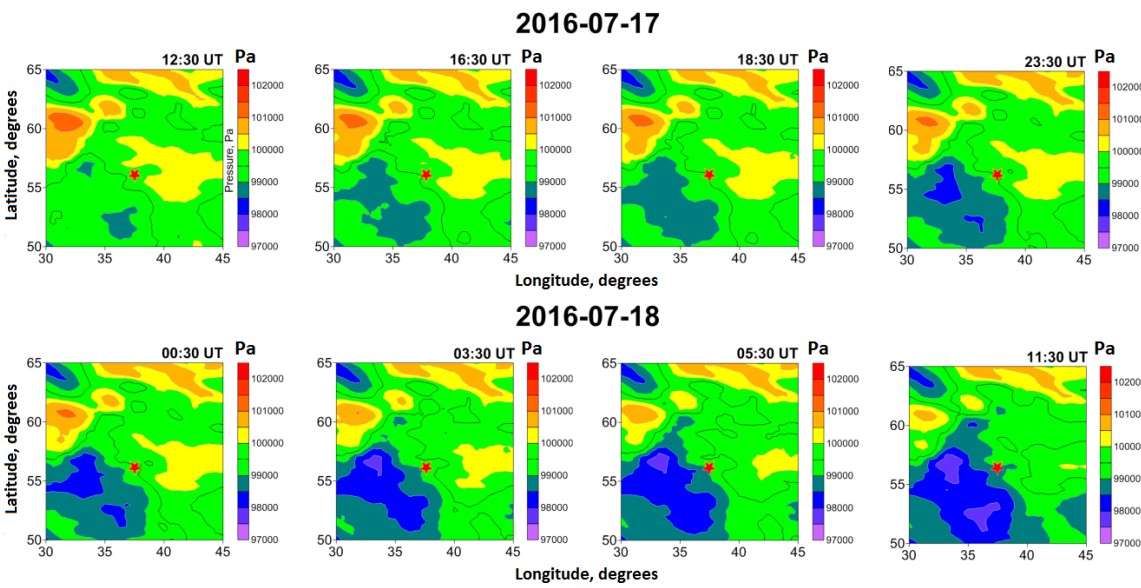

**Figure 3.** Observation data of surface pressure on July 18, 2016, available in the MERRA-2 database (https://disc.gsfc.nasa.gov/).

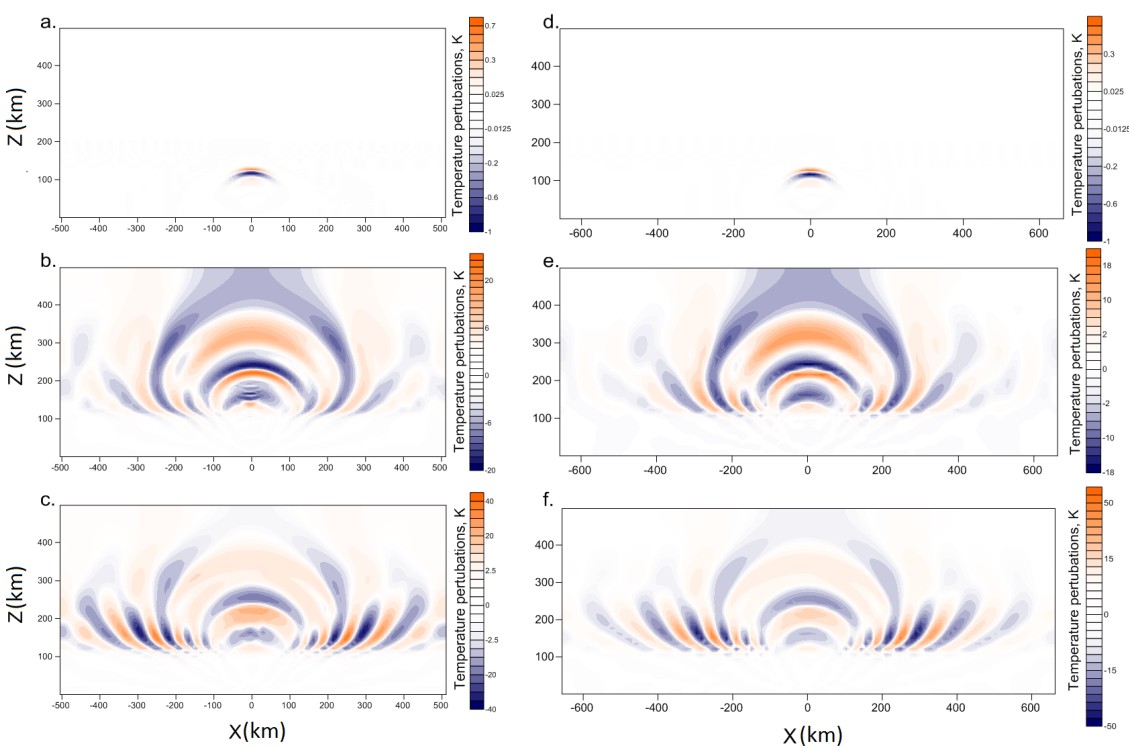

**Figure 4.** The plane y = 0 of cross section of the temperature field at a) t = 7 minutes, b) t = 40 minutes, c) t = 55 minutes for the computational area horizontally 1020 km × 1020 km and the the y = 0 cross section of the temperature field at d) t = 7 minutes, e) t = 40 minutes, f) t = 55 minutes for the computational area horizontally 1320 km × 1320 km.

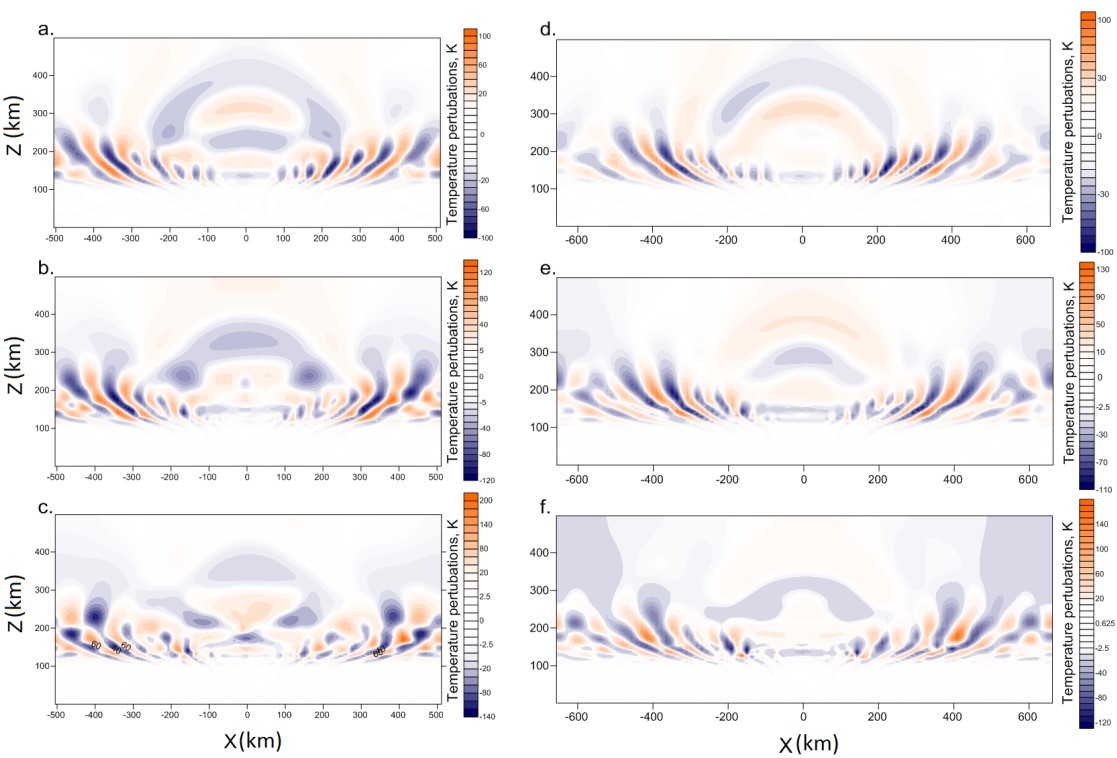

**Figure 5.** The plane y = 0 of cross section of the temperature field at a) t = 1 hour 30 minutes, b) t = 2 hours, c) t = 2 hours 30 minutes for the area of 1020km×1020km, and the plane y = 0 cross section of temperature field at d) t = 1 hour 30 minutes; e) t = 2 hours; f) t = 2 hours 30 minutes for the area of 1320km×1320km.