# Peer review of "PROPAGATION TO THE UPPER ATMOSPHERE OF ACOUSTIC-GRAVITY WAVES FROM ATMOSPHERIC FRONTS IN THE MOSCOW REGION"

_Annales Geophysicae, 2019_

## Referee Comment (RC1) · Anonymous Referee #1 · 4 Mar 2019

The paper is devoted to modeling and studying the propagation of acoustic-gravity waves in the atmosphere from pressure variations on the Earth's surface. As far as I know, the problem of wave propagation from pressure variations on a surface was not previously solved, but also was not even mathematically posed. The correct formulation of such a problem was formulated and proved in the previous recent works of the authors. Thus, the paper contains a new idea. Consideration of such a problem seems to be expedient since in modeling the wave propagation from tropospheric sources we encounter an obvious difficulty: detailed experimental information about tropospheric sources is usually lacking due to very complex spatial and temporal behavior of these sources. At the same time, these tropospheric sources lead to wave

pressure oscillations at the surface of the Earth, which are relatively easily recorded, and this experimental information can be used in simulations of atmospheric wave processes. The paper analyzes the observations of pressure variations for 2016 in the Moscow region. The case of extreme pressure variations is selected. For this event, the problem of vertical wave propagation is solved, which allows estimating in 200 K the temperature amplitude of the generated waves in the considered extreme event. The paper also contains an estimate of the amplitude of acoustic-gravity waves in the upper atmosphere, generated under calm meteorological conditions. In my opinion, the work is of interest to the journal and can be published after minor changes. I think, It would be good to check in detail the mathematical formulas, of which there are many, and more carefully check English. There are also the following specific comments:

Page 2 line 12: What does the word "development" mean in this context?

Page 2 line. 29: It is written 10-4, probably $10^{-4}$ actually.

Page 4 line 10: There is no explanation of $Q_{viscous}$.

Page 6 line 8: Instead of T must be $\tau$?

Page 5, line 23: What is meant by "input boundary source"?

Page 6 line 23 The text states "In Fig.4b, d, the wave field after 40 minutes and in Fig.4c, e 55 minutes". However, in the caption to the figure, 40 minutes corresponds to the image b, e, and 55 minutes - c, f. Correct it.

Please also note the supplement to this comment:
https://www.ann-geophys-discuss.net/angeo-2019-16/angeo-2019-16-RC1-supplement.pdf

---

## Short Comment (SC1) · Anonymous Referee 2 · 15 Mar 2019

Review Title: PROPAGATION TO THE UPPER ATMOSPHERE OF ACOUSTIC-GRAVITY WAVES FROM ATMOSPHERIC FRONTS IN THE MOSCOW REGION Authors: Yuliya Kurdyaeva, Sergey Kulichkov, Sergey Kshevetskii, Olga Borchevkina, and Elena Golikova

In this paper numerical simulation of acoustic-gravity wave (AGW) propagation up to the upper atmosphere is considered. The authors use their experimental data (4 microbarographs) in order to construct a source for the AGW generation. As a result of the simulations they were able to estimate the amplitude of temperature disturbances caused by the wave propagation at different altitudes, and to show that individual sources, measured by microbarographs, manifests itself at far distance as a single point source.

This paper appears to be suitable for publication with minor modifications which have to be done before the final acceptance. The reviewer has NOT done a detail check of all the math but it appears reasonable, while many gramma and typing errors were detected.

Major comment

The authors wrote that "A numerical model of wave propagation from pressure variations on the Earth's surface was developed in Kurdyaeva et al. (2018) ", and that "The study showed that the variable pressure on the Earth's surface uniquely determines the wave pattern, but this wave picture does not depend on the details of the temperature and density behavior on the Earth's surface (Kurdyaeva et al. (2018))." Therefore there is need to formulate the difference between the earlier works and present work in order to understand which points are new in the manuscript.

Minor comments

Line 29, page 2: 10-4 must be replaced by 10ˆ4

Line 31, page 2: there must be reference to the site with data utilized by the authors

Line 6, page 3 What does this mean "various other wave sources"

Lines 10, 11, page 5 Is it justified to include Coriolis force here for AGW?

Line 28, page 7 What is T there?

Line 4, page 7 Is this a really big difference taking 1020 km $\times$ 1020 km and 1320 km $\times$ 1320 km. for computational regions?

Capture of fig.4 and fig.5 What does this mean "The plane x = 0 of cross section..." if

one uses horizontal x axis and vertical z axis?

Line 10, page 9 Better to change "for the smaller region" by "for the region 1020 km x 1020 km"

Line 15, page 9 What is "domain" here? If it is 1020 km $\times$ 1020 km and 1320 km $\times$ 1320 km then should be mentioned.

Also I strongly recommend to the authors somehow to improve English before publication (may be to show the text to any professional translator).

---

## Author Comment (AC2) · 22 Mar 2019

Thanks for the very valuable comments. The manuscript is devoted to the study of the influence of acoustic-gravity waves (AGWs) from a moving atmospheric front on the state of the upper atmosphere. While in the work (Kurdyaeva et al. (2018)) the mathematical formulation of the problem on the propagation of waves from pressure variations at the lower boundary is investigated (the problem correctness is investigated) and the results of test calculations are given. The work (Kurdyaeva et al. (2018)) is not tied to any specific event in the atmosphere, it is essentially a mathematical work. In the present work, estimates of the amplitude of temperature disturbances in the upper

atmosphere caused by acoustic-gravity waves from the atmospheric front are given. This evaluation is performed for the first time.

Line 29, page 2: 10-4 must be replaced by 10ˆ4. It's corrected.

Line 31, page 2: This data is not publicly available.

Line 6, page 3: The change in atmospheric pressure can be associated with the activity of meteorological sources. However, it is impossible to completely exclude the effects on the wave pattern in the upper atmosphere of other wave sources unrelated to the meteorological events under consideration, for example, of sources of oscillations that are anthropogenic in nature. The study of wave propagation from the observed extreme pressure fluctuations allows hoping that the possible undesirable for our purposes influence of various other wave sources on the wave pattern is leveled.

Lines 10, 11, page 5: Yes, the Coriolis force terms can be ignored. It's correccted, these terms are omitted.

Line 28, page 7: T is temperature

Line 4, page 7: The dimensions of the considered areas in the horizontal dimension differ by 30%. The difference is not so great, but a more significant difference in the regions sizes s greatly increases the simulation time, because the dependence of the simulation time on the horizontal region scale is quadratic. At the same time, this difference between regions is significant, which is noticeable at large times, and that allows to notice that at not very long times, the wave picture and the main characteristics of the wave process weakly depend on the area size.

Capture of fig.4 and fig.5: Thank you. Indeed, there should be "The plane y = 0 of cross section ..." It's correxted.

Line 10, page 9: It's correxted.

Line 15, page 9: We believe that we just need to replace the "the" before the word

domain with the "a" and the sentence content will become clear.

Also, I strongly recommend to the authors somehow to improve English before publication (may be to show the text to any professional translator).

Thanks for the advice. We will do that.

The authors.

---

## Referee Comment (RC2) · Anonymous Referee #2 · 19 Apr 2019

This work aims to determine the vertical propagation of acoustic-gravity waves due to surface pressure variations on the upper atmosphere. The work uses microbarograph observations of pressure as well as MERRA reanalysis data as inputs into a 3D model. The 3D model then simulates the corresponding perturbations in the atmosphere. Unfortunately, this manuscript does not show that the goal is achieved. The use of observations and MERRA reanalysis is good. However, it isn't clearly proved that these pressure variations are due to acoustic-gravity waves. Also, the results section do not give a more physics-based description of the corresponding temperature perturbations. For example, why do the perturbations solely start above 100 km? What do we know of the propagation conditions for acoustic-gravity waves and why do we say that these

perturbations are indeed characteristic of acoustic-gravity waves? If the authors can clearly prove that these pressure variations and temperautre perturbations are indeed due to acoustic-gravity waves, then this paper is definitely worth publishing. Otherwise, this paper appears to solely be determining the upper atmosphere temperature perturbation of pressure changes in the surface. This isn't really a publishable result. As for the quality of writing, this manuscript needs a lot of grammar correction. The sentence constructions also need improvement.

Page 1, Line 13: Change 'medium state' to 'state of the atmosphere'.

Page 1, Lines 14-15: maximum of efficiency of what?

Page 1, Line 15: Add 'and' before 'mesoscale turbulence'.

Page 1, Line 19: Replace 'Dissipation of waves reached the upper atmosphere' to 'AGWs dissipating in the upper atmosphere'.

Page 1, Line 19: Repalce 'Atmospheric waves' with AGWs.

Page 2, Lines 2-4: I don't understand the long sentence "So, the authors noted in (Miller (1999), Fovell et al. (1992), Snively and Pasko (2003)) that meteorological sources excite short-period acoustic-gravity waves; the spectrum and the space-time pattern of the simulated wave process are in good agreement with the observations."

Page 2, Line 8: Remove 'layers of the'.

Page 2, Lines 10 - 12: The sentence "Phase transitions of water in the atmosphere are accompanied by the release/absorption of heat during the formation and evolution of clouds and alter the atmospheric pressure." seems out of place. Consider removing.

Page 2, Lines 17 - 19: I think temperature and density does affect the wave picture.

Page 3, Line 2: Change 'data of a real experiment' to 'observational data'.

Page 4 Lines 3 - 11: Describe briefly each equation.

Figure 1: Include a latitude-longitude axis on the figure.

Figure 2-3: Perhaps plot also the pressure variations from MERRA over the time-series in figure 2 for easier comparison.

Figure 4-5: Change z-axis to kilometers.

---

## Short Comment (SC2) · Anonymous Referee 2 · 19 Apr 2019

I think the author's reply demonstrates that they accepted all my comments. So I may recommend the manuscript for publication.

---

## Author Response (AR1)

**Received and published: 4 March 2019**

The paper is devoted to modeling and studying the propagation of acoustic-gravity waves in the atmosphere from pressure variations on the Earth's surface. As far as I know, the problem of wave propagation from pressure variations on a surface was not previously solved, but also was not even mathematically posed. The correct formulation of such a problem was formulated and proved in the previous recent works of the authors. Thus, the paper contains a new idea. Consideration of such a problem seems to be expedient since in modeling the wave propagation from tropospheric sources we encounter an obvious difficulty: detailed experimental information about tropospheric sources is usually lacking due to very complex spatial and temporal behavior of these sources. At the same time, these tropospheric sources lead to wave pressure oscillations at the surface of the Earth, which are relatively easily recorded, and this experimental information can be used in simulations of atmospheric wave processes. The paper analyzes the observations of pressure variations for 2016 in the Moscow region. The case of extreme pressure variations is selected. For this event, the problem of vertical wave propagation is solved, which allows estimating in 200 K the temperature amplitude of the generated waves in the considered extreme event. The paper also contains an estimate of the amplitude of acoustic-gravity waves in the upper atmosphere, generated under calm meteorological conditions. In my opinion, the work is of interest to the journal and can be published after minor changes. I think, It would be good to check in detail the mathematical formulas, of which there are many, and more carefully check English. There are also the following specific comments:

Many thanks to the distinguished reviewer for his careful work with the manuscript and very useful comments.

Below, our responses to the reviewer's comments are given.

*I think, It would be good to check in detail the mathematical formulas, of which there are many, and more carefully check English.*

Really, English needs some improvement. We will do it. Mathematical formulas will be carefully checked as well.

Page 2 line 12: What does the word "development" mean in this context?

It means formation and evolution. We will replace "development" so that no further questions arise.

Page 2 line. 29: It is written 10-4, probably 10-4 actually.

It will be corrected.

Page 4 line 10: There is no explanation of Q\_viscous.

 $Q_{viscous}$  is the force of viscous friction.

Page 6 line 8: Instead of T must be  $\tau$ ?

Indeed, T is  $\tau$  in meaning of the characteristic time of switching on of the boundary source. The slow switching on of the boundary source is introduced to eliminate transients. It will be corrected.

Page 5, line 23: What is meant by "input boundary source"?

**This is a misunderstanding. It will be corrected.**

Page 6 line 23

The text states "In Fig.4b, d, the wave field after 40 minutes and in Fig.4c, e 55 minutes". However, in the caption to the figure, 40 minutes corresponds to the image b, e, and 55 minutes - c, f. Correct it.

Thanks. It will be corrected.

The authors.

**Anonymous Referee 2**

**Received and published: 15 March 2019**

In this paper numerical simulation of acoustic-gravity wave (AGW) propagation up to the upper atmosphere is considered. The authors use their experimental data (4 microbarographs) in order to construct a source for the AGW generation. As a result of the simulations they were able to estimate the amplitude of temperature disturbances caused by the wave propagation at different altitudes, and to show that individual sources, measured by microbarographs, manifests itself at far distance as a single point source. This paper appears to be suitable for publication with minor modifications which have to be done before the final acceptance. The reviewer has NOT done a detail check of all the math but it appears reasonable, while many gramma and typing errors were detected

**Major comment**

The authors wrote that "A numerical model of wave propagation from pressure variations on the Earth's surface was developed in Kurdyaeva et al. (2018) ", and that "The study showed that the variable pressure on the Earth's surface uniquely determines the wave pattern, but this wave picture does not depend on the details of the temperature and density behavior on the Earth's surface (Kurdyaeva et al. (2018))." Therefore, there is need to formulate the difference between the earlier works and present work to understand which points are new in the manuscript.

Thanks for the very valuable comments. The manuscript is devoted to the study of the influence of acoustic-gravity waves (AGWs) from a moving atmospheric front on the state of the upper atmosphere. While in the work (Kurdyaeva et al. (2018)) the mathematical formulation of the problem on the propagation of waves from pressure variations at the lower boundary is investigated (the problem correctness is investigated) and the results of test calculations are given. The work (Kurdyaeva et al. (2018)) is not tied to any specific event in the atmosphere, it is essentially a mathematical work. In the present work, estimates of the amplitude of temperature disturbances in the upper atmosphere caused by acoustic-gravity waves from the atmospheric front are given. This evaluation is performed for the first time.

Line 29, page 2:10-4 must be replaced by 10-4

It's corrected.

Line 31, page 2: there must be reference to the site with data utilized by the authors

**This data is not publicly available.**

Line 6, page 3: What does this mean "various other wave sources"

The change in atmospheric pressure can be associated with the activity of meteorological sources. However, it is impossible to completely exclude the effects on the wave pattern in the upper atmosphere of other wave sources unrelated to the meteorological events under consideration, for example, of sources of oscillations that are anthropogenic in nature. The study of wave propagation from the observed extreme pressure fluctuations allows hoping that the possible undesirable for our purposes influence of various other wave sources on the wave pattern is leveled.

Lines 10, 11, page 5: Is it justified to include Coriolis force here for AGW?

Yes, the Coriolis force terms can be ignored. It's correccted, these terms are omitted.

Line 28, page 7: What is T there?

T is temperature.

Line 4, page 7: Is this a really big difference taking 1020 km  $\times$  1020 km and 1320 km  $\times$  1320 km for computational regions?

The dimensions of the considered areas in the horizontal dimension differ by 30%. The difference is not so great, but a more significant difference in the regions sizes s greatly increases the simulation time, because the dependence of the simulation time on the horizontal region scale is quadratic. At the same time, this difference between regions is significant, which is noticeable at large times, and that allows to notice that at not very long times, the wave picture and the main characteristics of the wave process weakly depend on the area size.

Capture of fig.4 and fig.5: What does this mean "The plane x = 0 of cross section..." if one uses horizontal x axis?

Thank you. Indeed, there should be "The plane y = 0 of cross section ..." It's corrected.

Line 10, page 9: Better to change "for the smaller region" by "for the region 1020 km x 1020 km"

It's correxted.

Line 15, page 9: What is "domain" here? If it is 1020 km  $\times$  1020 km and 1320 km  $\times$  1320 km then should be mentioned.

We believe that we just need to replace the "the" before the word domain with the "a" and the sentence content will become clear.

Also, I strongly recommend to the authors somehow to improve English before publication (may be to show the text to any professional translator).

Thanks for the advice. We will do that.

The authors.

**Anonymous Referee #2**

Received and published: 19 April 2019

This work aims to determine the vertical propagation of acoustic-gravity waves due to surface pressure variations on the upper atmosphere. The work uses microbarograph observations of pressure as well as MERRA reanalysis data as inputs into a 3D model. The 3D model then simulates the corresponding perturbations in the atmosphere. Unfortunately, this manuscript does not show that the goal is achieved.

The use of observations and MERRA reanalysis is good. However, it isn't clearly proved that these pressure variations are due to acoustic-gravity waves. Also, the results section do not give a more physics-based description of the corresponding temperature perturbations. For example, why do the perturbations solely start above 100 km? What do we know of the propagation conditions for acoustic-gravity waves? If the authors can clearly prove that these pressure variations and temperature perturbations are indeed due to acoustic-gravity waves, then this paper is definitely worth publishing. Otherwise, this paper appears to solely be determining the upper atmosphere temperature perturbation of pressure changes in the surface. This isn't really a publishable result. As for the quality of writing, this manuscript need a lot of grammar correction. The sentence constructions also need improvement.

**FIXED 19 April 2019:** I think the author's reply demonstrates that they accepted all my comments. So I may recommend the manuscript for publication.

Dear reviewer, thanks for your review. We hope that the answers to all the questions really satisfy you. Also thanks for the attentive attitude to the text. All comments and corrections will be taken into account.

**PROPAGATION TO THE UPPER ATMOSPHERE OF ACOUSTIC-GRAVITY WAVES FROM ATMOSPHERIC FRONTS IN THE MOSCOW REGION**

Yuliya Kurdyaeva1,2, Sergey Kulichkov3,4, Sergey Kshevetskii1, Olga Borchevkina1,2, and Elena Golikova3

1I.Kant Baltic Federal University, Kaliningrad, Russia

[revised manuscript text omitted]
 u w}{\partial x} + \frac{\partial \rho v w}{\partial y} = \frac{\partial p}{\partial z} + \left(\frac{\partial^2}{\partial x^2} + \frac{\partial^2}{\partial y^2}\right) \zeta(z) v + \frac{\partial}{\partial z} \zeta(z) \frac{\partial}{\partial z} v, \end{aligned} \\ & \frac{\partial \rho w}{\partial t} + \frac{\partial \rho u w}{\partial x} + \frac{\partial \rho v w}{\partial y} + \frac{\partial \rho w^2}{\partial z} = -\frac{\partial p}{\partial z} - \rho g + \left(\frac{\partial^2}{\partial x^2} + \frac{\partial^2}{\partial y^2}\right) \zeta(z) w + \frac{\partial}{\partial z} \zeta(z) \frac{\partial}{\partial z} w, \end{aligned} \\ & \frac{1}{\gamma - 1} \left(\frac{\partial P}{\partial t} + \frac{\partial P u}{\partial x} + \frac{\partial P v}{\partial y} + \frac{\partial P w}{\partial z}\right) = -P \left(\frac{\partial u}{\partial x} + \frac{\partial v}{\partial y} + \frac{\partial w}{\partial z}\right) + \\ & + \left(\frac{\partial^2}{\partial x^2} + \frac{\partial^2}{\partial y^2}\right) \kappa(z) T + \frac{\partial}{\partial z} \kappa(z) \frac{\partial}{\partial z} T + Q_0(z) + Q_{viscous} \end{aligned} \\ & Q_0(z) = -\frac{\partial}{\partial z} \kappa(z) \frac{\partial}{\partial z} T_0(z), P = \frac{\rho}{\mu R T} \end{aligned}$$

where t is time;  $p, \rho, T$  are pressure, density, and temperature; Rg is the universal gas constant; x, y, z and u, v, w are the coordinates and velocity components, respectively;  $\gamma$  is the adiabatic constant;  $\mu$  is the molecular weight; g is the gravity acceleration;  $\zeta$  and  $\kappa$  are viscosity and thermal conductivity coefficients;  $T_0(z)$  is the background temperature profile.  $Q_{viscous}$

**15 is the force of viscous friction.**

The dependences of medium parameters (viscosity and thermal conductivity coefficients, background density, temperature and pressure) on altitude are taken from the empirical model of the atmosphere NRLMSISE-00 (Picone et al. (2002)). The vertical grid is uneven; the optimal vertical grid is constructed by the program based on the real medium stratification.

The formulation of the problem of wave generation by pressure variations at the lower boundary is as follows. The system of equations (1) describes wave propagation. The periodic conditions at the horizontal boundaries of the computational domain are applied:

Periodic boundary conditions have the following disadvantage:with this formulation of the problem, the wave leaving a computational domain, for example, through the left boundary of the computational domain, then enters through the right boundary. Nevertheless, in this case, the distance travelled by the wave gradually increases and the amplitude of the wave propagating from the source gradually decreases due to spherical divergence and, cylindrical divergence at lager timeframes.

- 5 Therefore, given sufficiently large dimensions of the computational domain, the influence of the finiteness of the computational domain on the wave characteristics of interest, such as wave frequencies, spatial scales, amplitude, is not strong. The size of the computational domain in the horizontal plane is chosen experimentally. In this paper, for comparison, simulations are performed for two different computational domains in horizontal of 1020 km × 1020 km and 1320 km × 1320 km and the results showed that periodic conditions do not have a strong effect on the wave parameters of interest due to large computational
- 10 domain sizes.

15

Since we are interested only in waves generated by pressure fluctuations at the Earth's surface, the initial conditions

$$u(x,z,t=0) = 0, \qquad w(x,z,t=0) = 0, \qquad v(x,z,t=0) = 0, \qquad \rho(x,z,t=0) = \rho_0(z), \qquad T(x,z,t=0) = T_0(z) \quad (4)$$

correspond to the wave absence at the initial moment in time.

The upper boundary is at the altitude of h=500 km, and the upper boundary conditions are traditional for models of the thermosphere:

$$\frac{\partial T}{\partial z}\left(x,y,z=h,t\right)=0,\qquad \frac{\partial u}{\partial z}\left(x,y,z=h,t\right)=0,\qquad \frac{\partial v}{\partial z}\left(x,y,z=h,t\right)=0,\qquad w\left(x,y,z=h,t\right)=0.$$
(5)

[revised manuscript text omitted]